# FairOpt-PFN: Amortized Counterfactual Fairness with Optimal Fair Targets

**Enes Hasani** [1]  **Jake Robertson** [2 3 1]  **Frank Hutter** [2 3 1]

## Abstract

Ensuring algorithmic fairness is critical for both ethical and legal reasons. Grounded in causal modeling, Counterfactual Fairness provides a framework that aligns well with human intuition. However, traditional approaches to counterfactual fairness require knowledge of the true Structural Causal Model (SCM) at inference time. While much recent work has focused on removing this barrier, Prior-Data Fitted Networks (PFNs) have emerged as a particularly powerful paradigm for amortized bayesian inference. Recent work leverages PFNs to perform counterfactually fair inference using purely observational data, thereby bypassing the need for explicit causal modeling of the fairness task at hand. Yet, the pre-training objective utilized in this approach unnecessarily discards valid, counterfactually fair predictive signal. In this work, we propose a novel pre-training objective derived around utility-optimal decision-making subject to the counterfactual fairness constraint. Evaluating our approach through strictly controlled causal experiments that systematically scale bias, we demonstrate that this pre-training target yields a better fairness-utility trade-off.

## 1. Introduction

Algorithmic fairness is of fundamental importance in machine learning, motivated by both ethical principles and critical legal compliance (Barocas & Selbst, 2016). To better understand how biases affect decisions, causal fairness has recently emerged as a powerful framework that uses causal modeling to explicitly trace the pathways of discrimination (Kusner et al., 2017). Within this space, Counterfactual Fairness provides an intuitive standard, requiring a predictor's outcome for an individual to remain identical had their protected attribute been different, all else being equal (Kusner et al., 2017). However, constructing counterfactually fair predictors according to the algorithm suggested by Kusner et al. (2017) requires explicit knowledge of the true Structural Causal Model (SCM) at inference time. This is a practical limitation of the proposed algorithm, as gaining access to the ground truth causal model is difficult.

Prior-Data Fitted Networks (PFNs) have emerged as a powerful paradigm for amortized Bayesian inference (Müller et al., 2022). Recently, this approach has yielded highly successful foundation models for tabular data (Hollmann et al., 2025; Qu et al., 2026). Other works have successfully used this paradigm for causal discovery and inference tasks (Lorch et al., 2022; Robertson et al., 2025b; Ma et al., 2025; Balazadeh et al., 2025). Crucially, recent work shows that PFNs can perform counterfactually fair inference using observational data alone, bypassing explicit causal modeling at test time (Robertson et al., 2025a). However, the current pre-training strategy utilized in this foundation model relies on sub-optimal target construction. By bluntly severing causal paths of the protected attribute during pre-training, the existing methodology unnecessarily discards valid, counterfactually fair information, leading to a suboptimal fairness-utility trade-off. A second limitation is evaluative: existing FairPFN benchmarks do not explicitly report the magnitude of label-level bias in observational data. Consequently, it is difficult to distinguish whether low discrimination reflects successful removal of the protected attribute's causal influence or simply weak bias in the underlying data-generating process.

We address both limitations:

- **Methodologically,** we derive an optimal pre-training objective grounded in statistical decision theory. Rather than altering the causal graph, we compute the pre-training target for each individual in the test set as the Bayes-optimal decision from the conditional distribution over non-descendants of the protected attribute.

- **Evaluatively,** we use controlled, ATE-calibrated synthetic SCM benchmarks in which the causal effect of the protected attribute is systematically varied and the discrimination rate of an unfair ground-truth oracle is reported as a diagnostic of label-level bias in observational data.

[1]University of Freiburg [2]Prior Labs [3]ELLIS Institute Tübingen. Correspondence to: Enes Hasani <hasanie@informatik.uni-freiburg.de>.

*Proceedings of the $2^{nd}$ ICML Workshop on Foundation Models for Structured Data*, Seoul, South Korea. 2026. Copyright 2026 by the author(s).

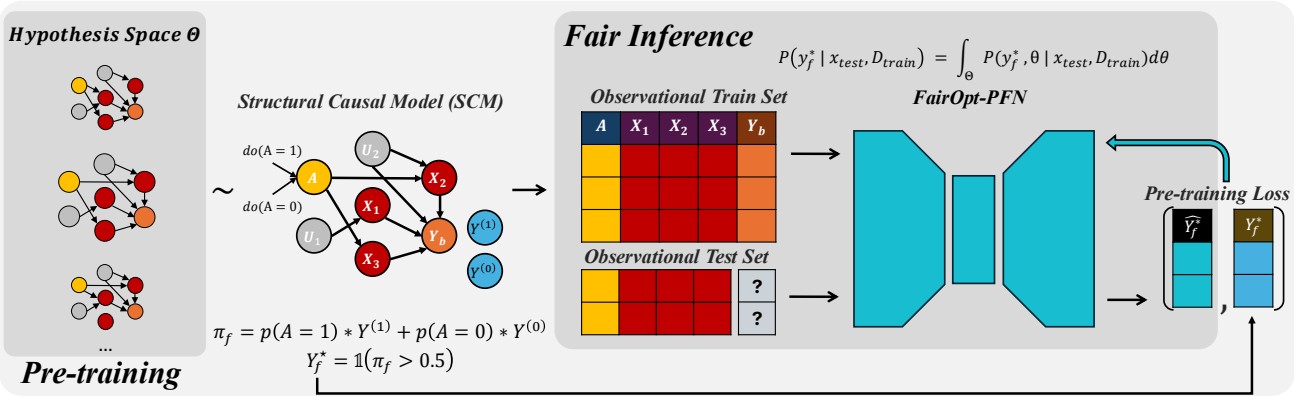

*Figure 1.* During pre-training, a Structural Causal Model (SCM) $\theta$ is sampled from the hypothesis space $\Theta$ to generate biased splits $\mathcal{D}_{\text{train}}$ and $\mathcal{D}_{\text{test}}$. The model utilizes $\mathcal{D}_{\text{train}}$ as context to infer the underlying hypothesis that generated the biased data. Guided by the pre-training loss, the model is optimized to predict the targets $y_f^*$ for the test points. A foundation model pre-trained in this manner can take a biased, real-world training set as context and make fair predictions on the biased test set.

## 2. Background

**Counterfactual Fairness.** A predictor is counterfactually fair if it gives the same prediction had the individual belonged to another demographic group while having all other background traits remain the same (Kusner et al., 2017). To achieve this invariance, Kusner et al. (2017) introduce what we will refer to as the *counterfactual fairness lemma*: a predictor is strictly counterfactually fair if its predictions depend solely on the non-descendants of the protected attribute $A$, denoted as $\text{ND}(A)$. This restriction structurally ensures that any causal pathways originating from the protected attribute cannot influence the downstream prediction.

**Prior-Data Fitted Networks (PFNs).** Prior-Data Fitted Networks (PFNs) are transformer architectures designed for implicit Bayesian inference (Müller et al., 2022). Pre-trained on synthetic datasets sampled from a prior distribution over a broad hypothesis space $\Theta$, when presented with a set of training data $\mathcal{D}_{\text{train}}$ and a new test query $x_{\text{test}}$, they approximate the posterior predictive distribution over the target $y$ by implicitly integrating over $\Theta$: $P(y \mid x_{\text{test}}, \mathcal{D}_{\text{train}}) = \int_{\Theta} P(y, \theta \mid x_{\text{test}}, \mathcal{D}_{\text{train}}) \, d\theta$.

**Causal Fairness with PFNs.** Robertson et al. (2025a) instantiate this hypothesis space $\Theta$ as a diverse set of Structural Causal Models (SCMs). Each of these SCMs models the data-generating process of a biased observational dataset. Concurrently, by removing the outgoing causal edges from the protected attribute $A$, for each individual in the test set these SCMs generate target nodes $y_f$ that model what the outcome would have been had there been no causal influence of the protected attribute. Consequently, by pre-training on these targets, FairPFN implicitly approximates the posterior predictive distribution over the fair target $y_f$: $P(y_f \mid x_{\text{test}}, \mathcal{D}_{\text{train}}) = \int_{\Theta} P(y_f, \theta \mid x_{\text{test}}, \mathcal{D}_{\text{train}}) \, d\theta$. While this approach structurally guarantees counterfactually

fair targets, permanently deleting edges acts as a blunt instrument that can aggressively destroy counterfactually fair task-relevant information. This leaves an open challenge: deriving a principled pre-training target that maximizes predictive utility while satisfying counterfactual fairness.

## 3. Methodology

**The Optimal Fair Target.** To guarantee counterfactual fairness within a given Structural Causal Model, the counterfactual fairness lemma dictates that predictions regarding a target variable $Y$ must rely exclusively on the non-descendants of the protected attributes, $\text{ND}(A)$. While this structural restriction ensures fairness, it still permits a vast space of valid predictive functions. A mathematically principled approach to isolating the optimal prediction from this space relies on statistical decision theory: establishing a predefined loss function $\mathcal{L}(y, \hat{y})$ and selecting the exact decision that minimizes the expected loss.

Combining these principles dictates the definition of the optimal fair target, $y_f^\star$. Within a given SCM, the optimal fair decision for any specific instance evaluates to the value that minimizes the expected loss over the true conditional distribution of the non-descendants:

$$y_f^\star = \arg\min_{\hat{y}} \mathbb{E}_{y \sim P(y \mid \text{ND}(A))}[\mathcal{L}(y, \hat{y})] \qquad (1)$$

Because this target is computed exclusively as a function of the non-descendants, it structurally satisfies counterfactual fairness while achieving the minimum expected loss permitted under that constraint.

**Optimal Fair Target for Binary Classification.** We utilize the standard 0-1 loss to optimize for classification accuracy in a binary setting under the counterfactual fairness constraint. Consequently, from (1), the optimal fair tar-

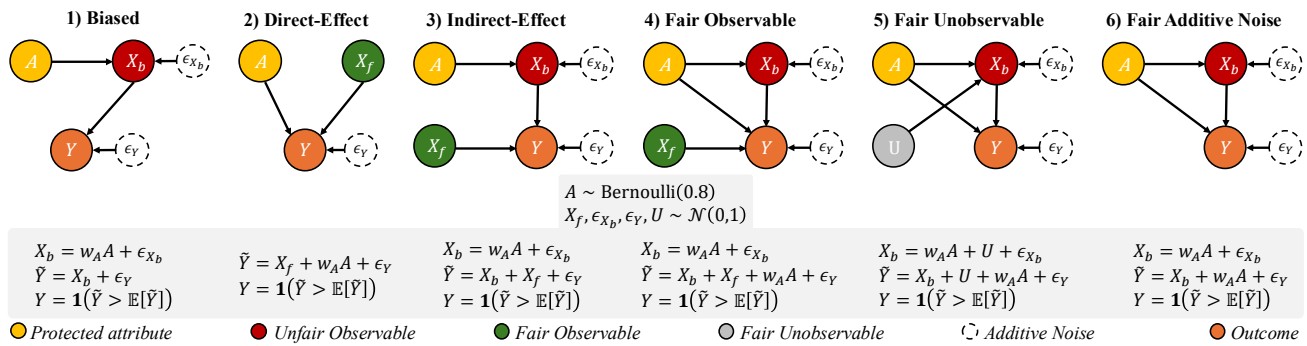

Figure 2. The hypothesis space $\Theta$. All causal edge weights are strictly set to 1, except for $w_A$ (edges originating from the sensitive attribute $A$). To model different bias magnitudes, $w_A$ is uniformly sampled from 31 precomputed values, explicitly calibrated to induce target Average Treatment Effect (ATE) of A on Y linearly spaced in the interval $[0, 0.99]$.

get $y_f^\star$ reduces to the mode of the conditional distribution: $y_f^\star = \arg\max_{\hat{y}\in\{0,1\}} P(y = \hat{y} \mid \text{ND}(A))$.

Let $y^{(1)}$ and $y^{(0)}$ denote the potential outcomes under the interventions $do(A = 1)$ and $do(A = 0)$, respectively. As detailed in Appendix A, we compute $y_f^\star$ as follows:

$$\pi_f = p(A = 1) \cdot y^{(1)} + p(A = 0) \cdot y^{(0)} \qquad (2)$$

$$y_f^\star = \mathbb{1}(\pi_f > 0.5) \qquad (3)$$

In essence, $y_f^\star$ is the most likely potential outcome for an individual.

**Pre-training for Fair Inference.** This optimal target integrates naturally into the PFN pre-training loop. During pre-training, SCMs are repeatedly sampled from the prior over $\Theta$ to generate biased observational splits $(\mathcal{D}_{\text{train}}, \mathcal{D}_{\text{test}})$. For each query $x_{\text{test}} \in \mathcal{D}_{\text{test}}$, the target $y_f^\star$ is computed using the sampled SCM's graphical structure. By optimizing the network to predict $y_f^\star$ from the biased context using the training loss, the PFN learns to implicitly approximate the posterior predictive distribution over the fair optimal target given $(x_{\text{test}}, \mathcal{D}_{\text{train}})$ as input: $P(y_f^\star \mid x_{\text{test}}, \mathcal{D}_{\text{train}}) = \int_\Theta P(y_f^\star, \theta \mid x_{\text{test}}, \mathcal{D}_{\text{train}}) \, d\theta$. The pre-training process is illustrated in Figure 1.

## 4. Experiments and Results

**Experimental Setup.** The hypothesis space $\Theta$ is instantiated as a set of six DAGs adapted from Robertson et al. (2025a), each modeling a different causal relationship between the protected attribute $A$, observed covariates $X$, and target label $Y$. Unlike previous work that relies on uncalibrated random sampling of causal weights and exogenous noise—which can inadvertently generate unbiased observational datasets where fair inference is trivial—we employ a strictly controlled sampling strategy. As detailed in Appendix A, we restrict the causal weight of the protected attribute $(w_A)$ to a precomputed, ATE-calibrated set of values. This ensures that the structural bias in our datasets

scales predictably, providing a systematic stress test for fair inference. Figure 2 shows the hypothesis space.

**Pre-Training.** We pre-train three distinct network variants[2], all sharing the lightweight architecture of Pfefferle et al. (2025): Nano-TabPFN (pre-trained conventionally on the observational label $y$), Nano-FairPFN (pre-trained on the edge-deleted target $y_f$), and our proposed Nano-FairOpt-PFN (pre-trained on the optimal target $y_f^\star$ generated using (2) and (3)). Full pre-training details are provided in Appendix A.

**Baselines.** Because the Posterior Predictive Distribution (PPD) is analytically tractable for our hypothesis space $\Theta$, we compute the exact Bayes-optimal reference predictors for each target. These baselines allow us to disentangle the fairness-utility trade-off attributable to the choice of fair target from the approximation error of learned PFNs. **Unfair Oracle Predictor** simply returns the ground-truth unfair observational label $y$ for each test query.

**Metrics.** Based on the definition of Counterfactual Fairness, we say that an individual is discriminated against if the predictor's outcome change under an intervention that changes the protected attribute while keeping other exogenous noise fixed. We define **Discrimination Rate** as the probability of an individual being discriminated against. The Discrimination Rate of the Unfair Oracle Predictor serves as an indicator of how biased the observational data is. **Inaccuracy** is evaluated as the expected 0-1 loss with respect to the true observational label $y$. Monte Carlo sampling of datasets from the prior over hypothesis space $\Theta$ was dynamically continued until the full width of the 95% confidence intervals for both metrics was at most 0.01. Details of the evaluation procedure are provided in Appendix B.

**Results.** The main finding is that the choice of targets during pre-training has a significant impact on the fairness-utility

---

[2]The codebase is available at https://github.com/EnesHasani/FairOpt-PFN

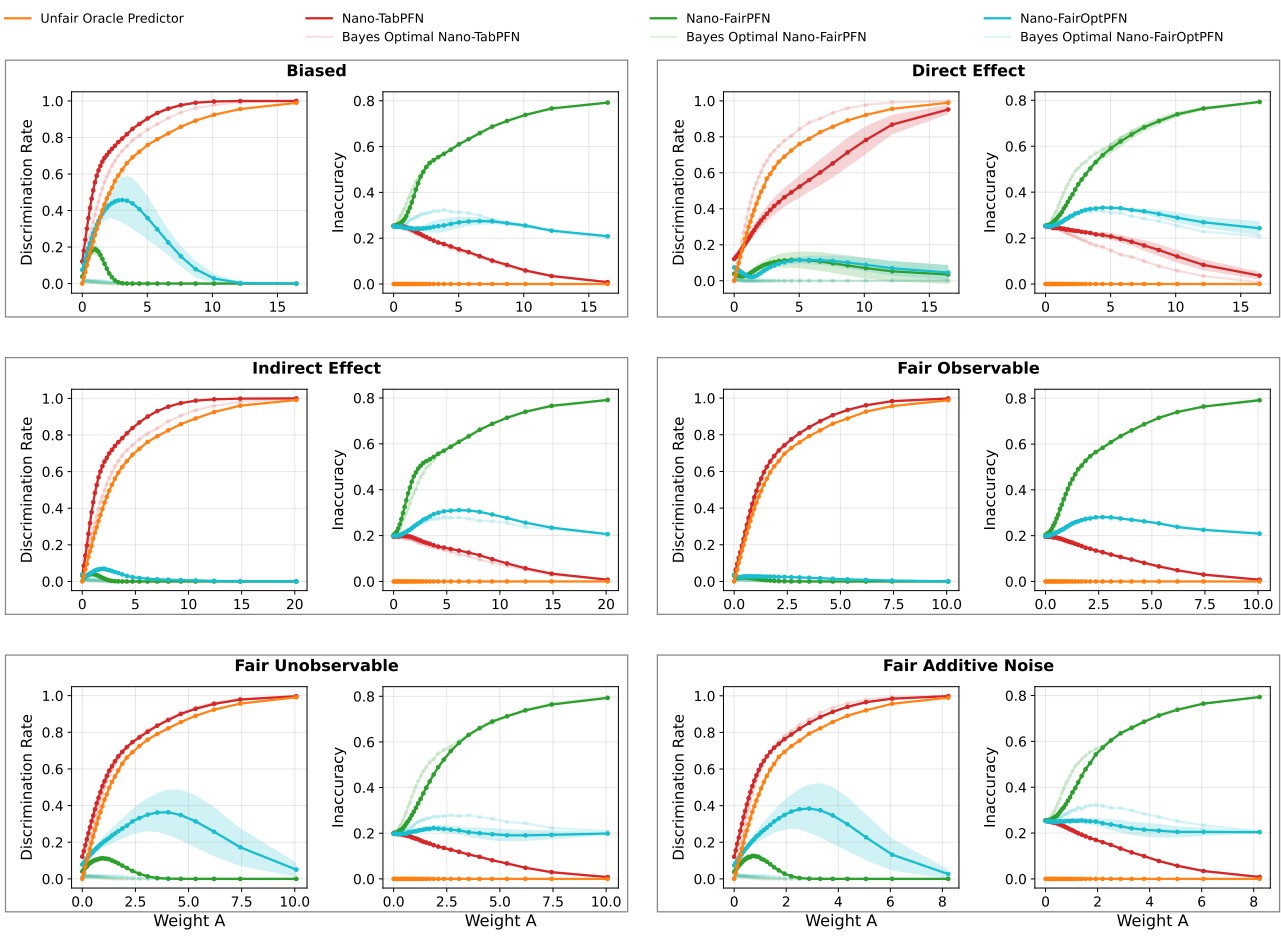

*Figure 3.* Each boxed panel corresponds to one DAG and reports discrimination rate (left) and inaccuracy (right) as a function of Weight A. Solid curves show the learned Nano-TabPFN, Nano-FairPFN, and Nano-FairOpt-PFN models; lighter curves show the corresponding Bayes-optimal references. The Discrimination Rate of the Unfair Oracle Predictor serves as an indicator of how biased the observational data is. For the learned models, shaded bands indicate confidence intervals computed from the standard error across five independently pretraining runs.

trade-off of the resulting PFNs. When the observational data is unbiased, fair inference is trivial and all models perform similarly. However, as the data becomes increasingly biased, Nano-FairOpt-PFN (blue) recovers much of the accuracy lost by Nano-FairPFN (green) while remaining significantly less discriminatory than Nano-TabPFN (red). These results are consistent across all six case studies.

Furthermore, evaluating against the exact analytical baselines isolates the source of the remaining bias. The Bayes-optimal references (faded lines) demonstrate that both fair pre-training targets theoretically admit no discrimination across our benchmarks. Therefore, the residual unfairness observed in the learned Nano-FairOpt-PFN—most visible as an increase in discrimination at lower causal weights—is a result of the transformer's empirical approximation error, rather than a failure of the pre-training strategy itself. For results in other hypothesis spaces, refer to Appendix C.

## 5. Discussion and Future Work

We formulated PFN pre-training as an optimal decision problem subject to counterfactual fairness constraint, addressing a key limitation in the optimization target used in Robertson et al. (2025a). Our controlled experiments demonstrate a better fairness-utility trade-off than FairPFN. Future work will scale this formulation (1) to multiple protected attributes and non-binary tasks, yielding a zero-shot model needing no inference-time causal graphs. Hyperparameter tuning will also improve PPD approximation and address learning bottlenecks. Finally, to accommodate legally permissible "business necessity" variables (Barocas & Selbst, 2016), we will integrate path-specific counterfactual fairness (Chiappa, 2019; Nabi & Shpitser, 2018; Plečko & Bareinboim, 2024), selectively permitting causal influence through valid mediator pathways to further improve predictive utility while ensuring legal compliance.

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

## A. Pre-Training Details

We evaluate our approach across two types of hypothesis spaces. The *Generic* space encompasses all six DAG case studies, whereas a *Specific* space consists of only a single DAG. Models are evaluated on the same hypothesis space used during their pre-training. Structurally, all non-sensitive edge weights are fixed to 1, with variations applied exclusively to edges originating from the protected attribute. Because pre-training and the exact analytical baselines utilize the identical hypothesis space, any differences between the learned and Bayes-optimal predictors stem solely from the empirical approximation error of the respective Posterior Predictive Distribution (PPD).

**ATE-Calibrated Weight Sequence.** For each case study $c$ and Bernoulli parameter $p = P(A = 1)$, we do not sample the outgoing weight of $A$ from a continuous distribution. Instead, we precompute a list of 31 values of $w_A$ corresponding to target ATEs linearly spaced in $[0, 0.99]$. Let $\Phi$ denote the standard normal CDF. For the SCMs used in our generators, the induced ATE of $A$ on $Y$ takes the closed form:

$$\text{ATE}_c(w; p) = \Phi\left(\frac{m_c w(1-p)}{\sigma_c}\right) - \Phi\left(\frac{-m_c w p}{\sigma_c}\right), \quad (4)$$

where $m_c$ is the total number of directed unfair paths from $A$ to $Y$, and $\sigma_c$ is the standard deviation of the aggregated Gaussian noise. For every target value $\tau$, we solve $\text{ATE}_c(w; p) = \tau$ by one-dimensional root finding over $w \in [0, 50]$, round the solution to three decimals, and cache the resulting 31-value grid. The constants used in the implementation are listed in Table 1.

*Table 1.* Constants used to calibrate $w_A$ to target ATEs.

| Case study family | $\sigma_c$ | $m_c$ |
|---|---|---|
| Biased, Direct Effect | $\sqrt{2}$ | 1 |
| Indirect Effect | $\sqrt{3}$ | 1 |
| Fair Observable, Fair Unobservable | $\sqrt{3}$ | 2 |
| Fair Additive Noise | $\sqrt{2}$ | 2 |

**Prior Sampling and Training Targets.** During pre-training, one case study is sampled uniformly from the selected set and one calibrated weight $w_A$ is sampled uniformly from its corresponding 31-value grid. The dataset size $n$ is drawn log-uniformly from $[100, 1000]$, the train/test split is fixed to $80/20$, and all exogenous as well as additive noise variables are sampled from zero-mean Gaussians with unit standard deviation. The protected attribute is binary with user-specified Bernoulli parameter $p = P(A = 1)$. For SCMs with two outgoing edges from $A$, the same sampled value of $w_A$ is reused on both edges. Each generator returns an observational dataset together

with one counterfactual test set obtained by flipping $A$ while keeping the exogenous noise fixed.

From these draws, the training code constructs three supervision targets for the test set. Nano-TabPFN is trained on the observational label $y$. Nano-FairPFN is trained on the edge-deleted fair label $y_f$, obtained by setting all outgoing coefficients of $A$ to zero and regenerating the test labels. Nano-FairOpt-PFN is trained on the optimal fair label $y_f^\star$. For a binary classification task, we aim to compute the fair probability $\pi_f = P(y = 1 \mid \text{ND}(A))$ by marginalizing out the protected attribute $A$:

$$\pi_f = \sum_{a \in \{0,1\}} P(y = 1 \mid A = a, \text{ND}(A)) P(A = a \mid \text{ND}(A)) \quad (5)$$

In Counterfactual Fairness, $A$ is modeled as an exogenous node, hence the conditional probability $P(A = 1 \mid \text{ND}(A))$ simplifies to $p = P(A = 1)$. Second, because conditioning on $\text{ND}(A)$ inherently fixes the individual's exogenous background noise $U$, the outcome generation is fully deterministic once $A$ is known. Consequently, the conditional probability $P(y = 1 \mid A = a, \text{ND}(A))$ collapses to the individual's potential outcome, $y^{(a)} \in \{0, 1\}$, under the intervention $do(A = a)$.

Applying these two structural simplifications, the fair probability reduces to a weighted average of the potential outcomes:

$$\pi_f = p y^{(1)} + (1-p) y^{(0)} \quad (6)$$

The optimal fair target is then generated by thresholding this probability to find the mode, effectively selecting the most likely potential outcome for the individual:

$$y_f^\star = \mathbb{1}(\pi_f > 0.5) = \arg\max_{\hat{y} \in \{0,1\}} P(y = \hat{y} \mid \text{ND}(A)) \quad (7)$$

**Optimization.** All nano PFN variants use the same lightweight transformer backbone: 6 layers, embedding size 192, 6 attention heads, MLP hidden size 768, and 2 output logits. We optimize with cross-entropy loss and AdamW Schedule-Free using learning rate $10^{-4}$ and zero weight decay. Each epoch consists of 500 freshly sampled synthetic datasets, gradients are clipped to norm 1, and the trainer keeps the latest checkpoint together with the top three checkpoints ranked by training loss. The training script is configured for 40,000 epochs, but actual runs are terminated by a wall-clock budget; the experiment pipeline supports both case-study-specific pre-training and a generic model trained jointly on all six case studies. Generic models were trained for three hours, and specific models for five minutes. We report results aggregated over five independent pre-training runs across 14 distinct hypothesis spaces (the six specific case studies plus the single generic setting, each evaluated at two different values of $p$).

## B. Evaluation Metrics

To evaluate the counterfactual fairness of a model, we measure the discrepancy in its predictions under a counterfactual intervention. For an individual $i$ with observed features $x^{(i)}$ generated by exogenous noise $U^{(i)}$ and protected attribute $A^{(i)}$, the predictor outputs a hard prediction by taking the argmax of the posterior predictive distribution conditioned on a training context $\mathcal{D}_{\text{train}}$:

$$\hat{y}^{(i)} = \arg\max_y P(y \mid x^{(i)}, \mathcal{D}_{\text{train}}) \qquad (8)$$

We compute the counterfactual features $x_{CF}^{(i)}$ by intervening to flip the protected attribute ($A \leftarrow 1 - A^{(i)}$) while keeping the original exogenous noise $U^{(i)}$ strictly fixed. Passing this counterfactual through the predictor yields:

$$\hat{y}_{CF}^{(i)} = \arg\max_y P(y \mid x_{CF}^{(i)}, \mathcal{D}_{\text{train}}) \qquad (9)$$

Based on the definition of Counterfactual Fairness, we consider an individual to be discriminated against if the predictor's decision changes under this intervention (i.e., $\hat{y}^{(i)} \neq \hat{y}_{CF}^{(i)}$). To quantify the overall unfairness of a model, we define the *Discrimination Rate* as the probability of this discrepancy occurring for a randomly sampled pair: an individual $x^{(i)} = (x, U, A)$, and an in-context group of individuals $\mathcal{D}_{\text{train}}$:

$$\text{Discrimination Rate} = P_{\mathcal{D}_{\text{train}},(x,U,A)}(\hat{y} \neq \hat{y}_{CF}) \qquad (10)$$

We compute the empirical estimate of this metric, $\widehat{\text{Discrimination Rate}}$, via Monte Carlo simulation. We sample $M$ distinct pairs of $(\mathcal{D}_{\text{train}}, \mathcal{D}_{\text{test}})$, compute the discrimination rate for the $N$ individuals within each test set, and average the results across all datasets:

$$\widehat{\text{Discrimination Rate}} = \frac{1}{M}\sum_{j=1}^{M}\left(\frac{1}{N}\sum_{i=1}^{N}\mathbb{1}(\hat{y}^{(i,j)} \neq \hat{y}_{CF}^{(i,j)})\right) \qquad (11)$$

where $\hat{y}^{(i,j)}$ and $\hat{y}_{CF}^{(i,j)}$ represent the standard and counterfactual predictions for the $i$-th individual in the $j$-th sampled dataset.

For utility, we evaluate the predictive *Inaccuracy* (the expected 0-1 loss) with respect to the true observational label $y$. This is estimated analogously by averaging the test set inaccuracy across the $M$ sampled datasets:

$$\widehat{\text{Inaccuracy}} = \frac{1}{M}\sum_{j=1}^{M}\left(\frac{1}{N}\sum_{i=1}^{N}\mathbb{1}(\hat{y}^{(i,j)} \neq y^{(i,j)})\right) \qquad (12)$$

For every (DAG, Weight A) pair in our hypothesis space, we dynamically continued Monte Carlo sampling until the full width of the 95% confidence intervals for both metrics was at most 0.01.

## C. Additional Results

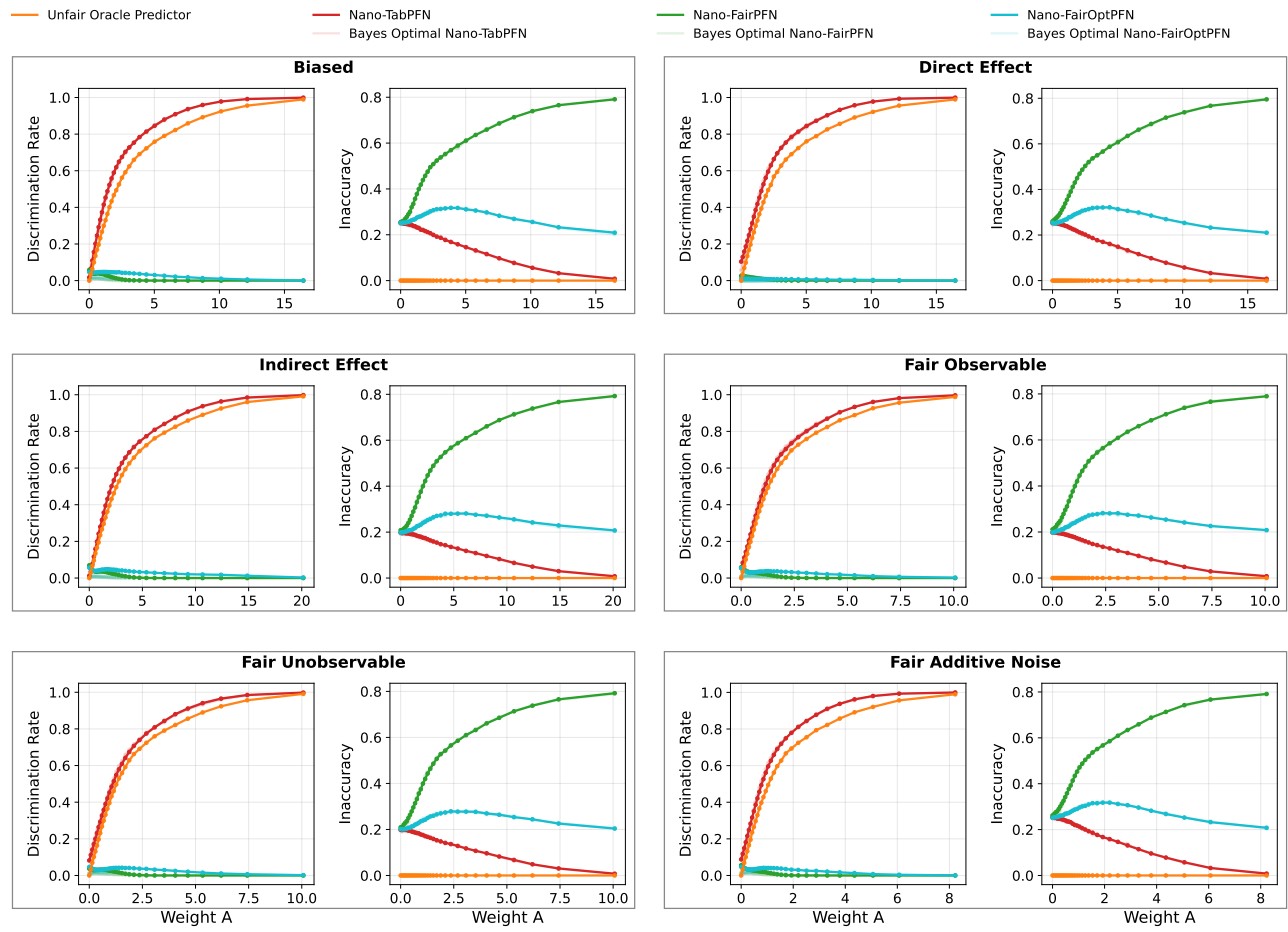

*Figure 4.* Specialized-pretraining results for $P(A = 1) = 0.8$. One model was pretrained on a fixed DAG and only the causal weights outgoing from A were sampled randomly. In total 6 six such models were pre-trained and evaluated, one for each case study. The task was easier compared to the generic setting. It can be seen the Bayes Optimal curves overlap almost perfectly with the curves of learned PFNs.

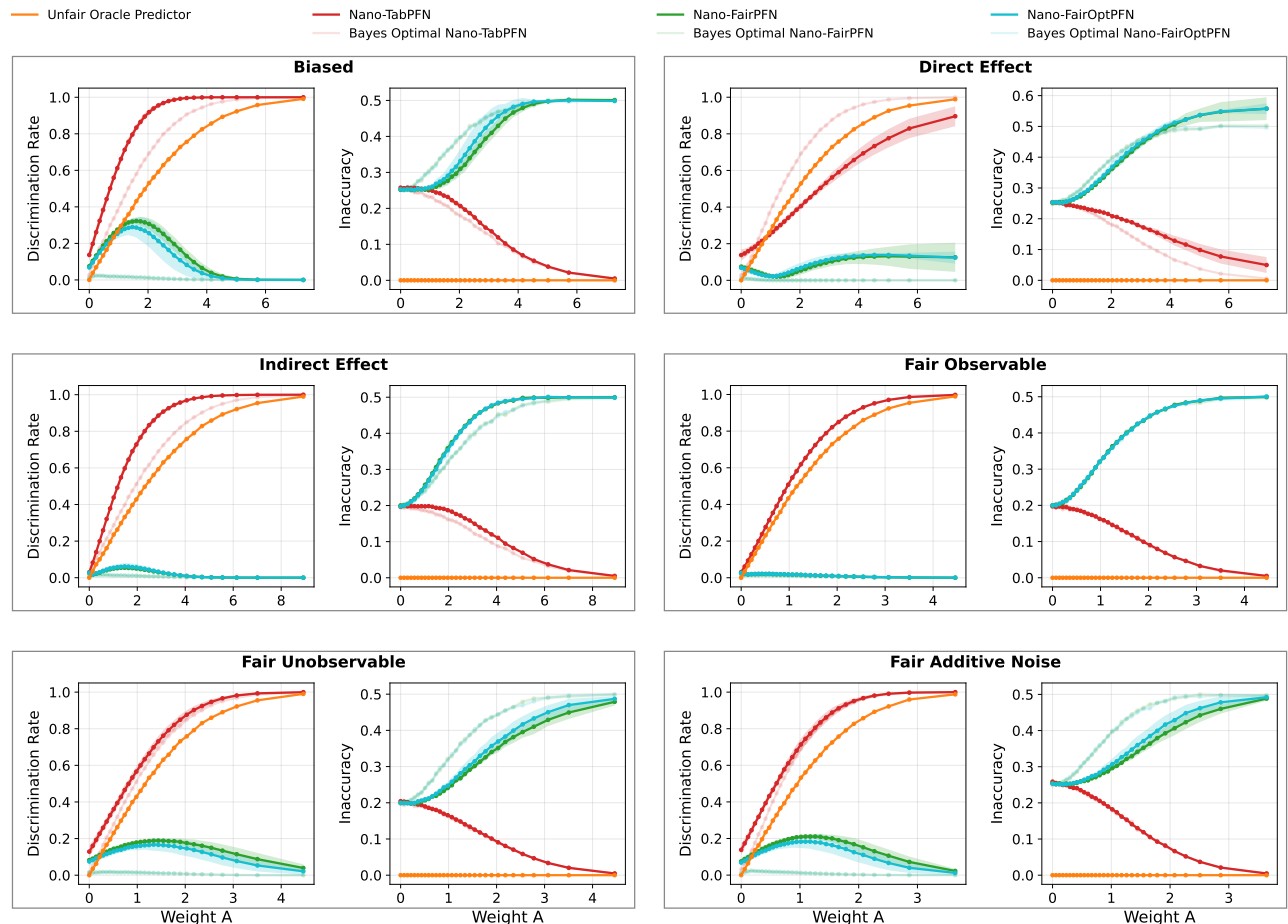

*Figure 5.* Additional results for $P(A = 1) = 0.5$. In this hypothesis space, $y_f^\star = y_f$ during pre-training, hence pre-training yields identical FairPFN and FairOptPFN. So, there are hypothesis spaces where FairPFN coincides with FairOptPFN, however that is not true in general. A foundation model can be thought as a model that was trained on a set of all these hypothesis spaces, and on this set FairOptPFN would yield better fairness-accuracy tradeoff.

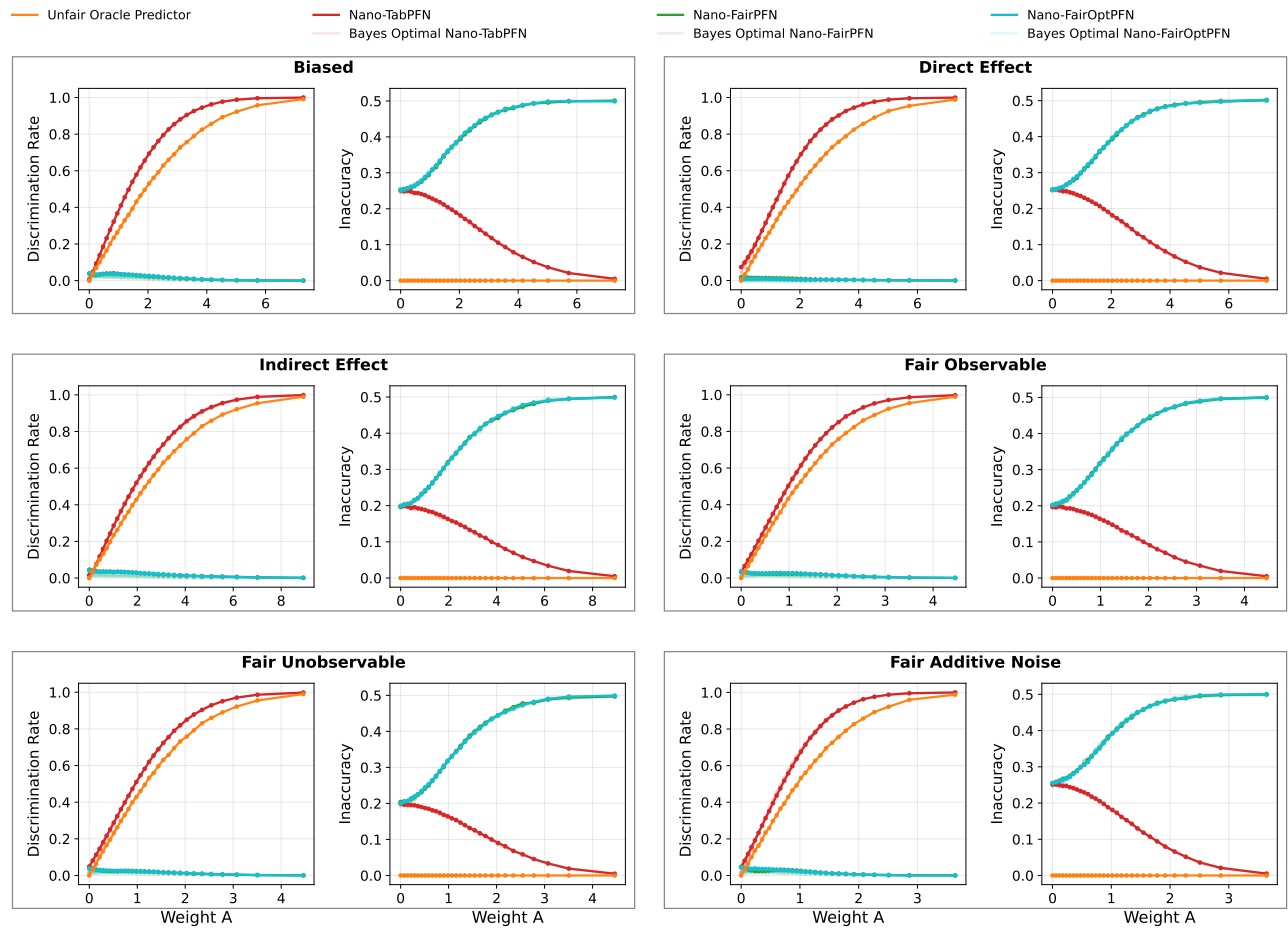

*Figure 6.* Specialized-pretraining results for $P(A = 1) = 0.5$. One model was pretrained on a fixed DAG and only the causal weights outgoing from A were sampled randomly. In total 6 six such models were pre-trained and evaluated, one for each case study. The task was easier compared to the generic setting. It can be seen the Bayes Optimal curves overlap almost perfectly with the curves of learned PFNs.

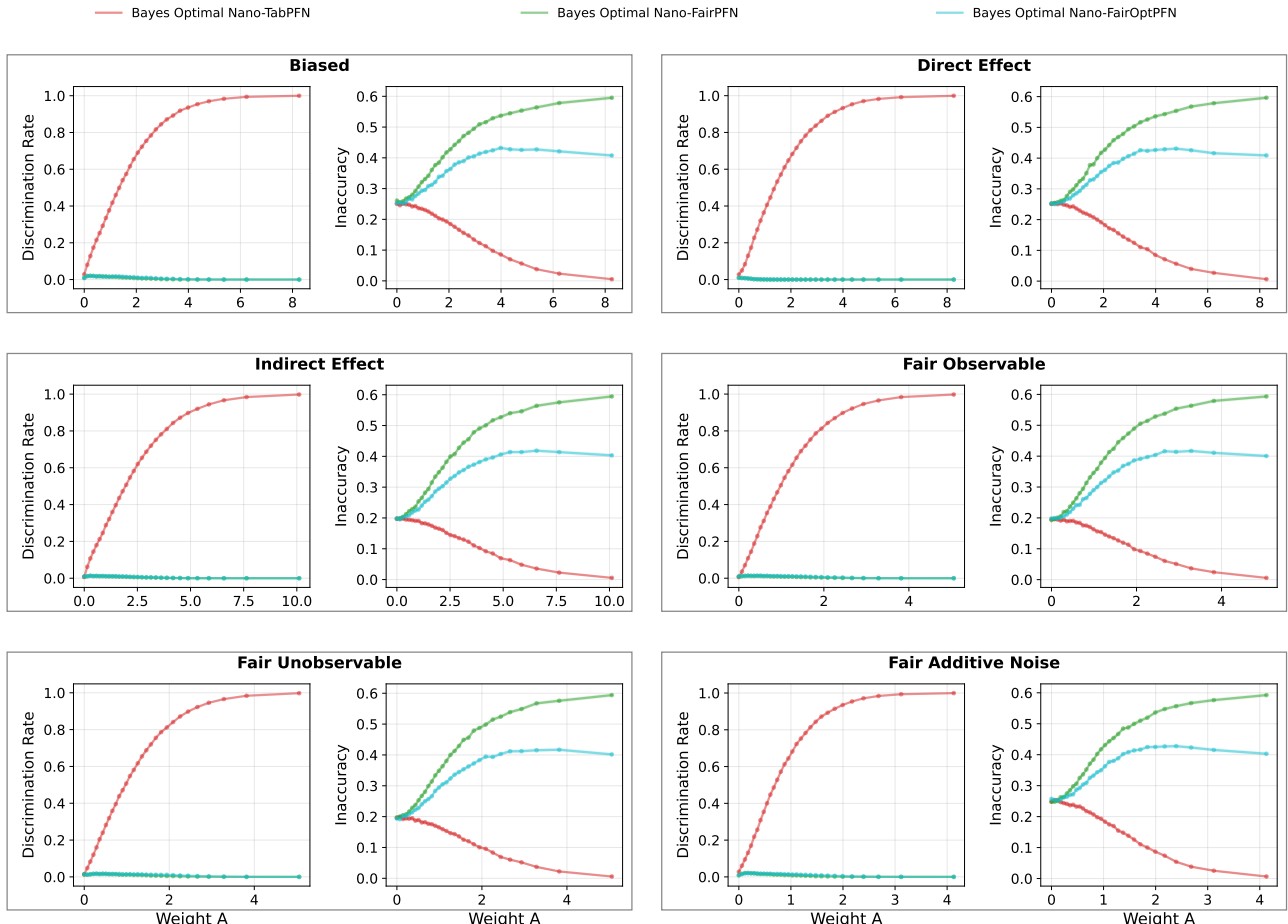

*Figure 7.* Additional results for $P(A = 1) = 0.6$. Because pre-training and evaluation of PFNs is expensive, we used these Bayes Optimal curves to show what the curves of learned PFNs would look like for this setting (had the learned PFNs approximated PPD with zero approximation error). When $\Theta$ is parametrized with a $P(A = 1) > 0.5$, then $y_f \neq y_f^\star$. In these hypothesis spaces FairOptPFN does better fairness-accuracy tradeoff.

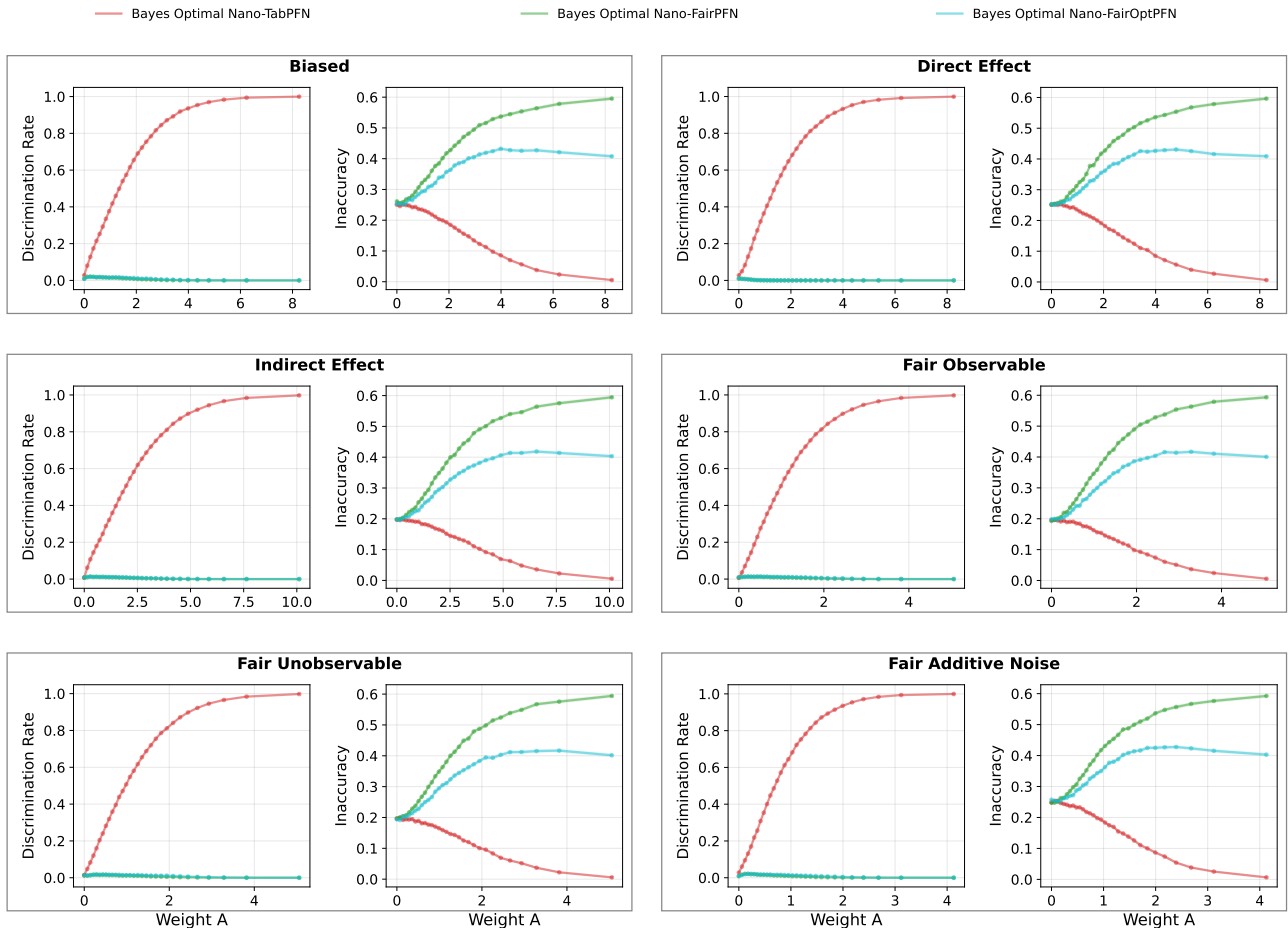

*Figure 8.* Specialized results for $P(A = 1) = 0.6$. Here the DAG is fixed, and only the weights exiting A are sampled randomly.

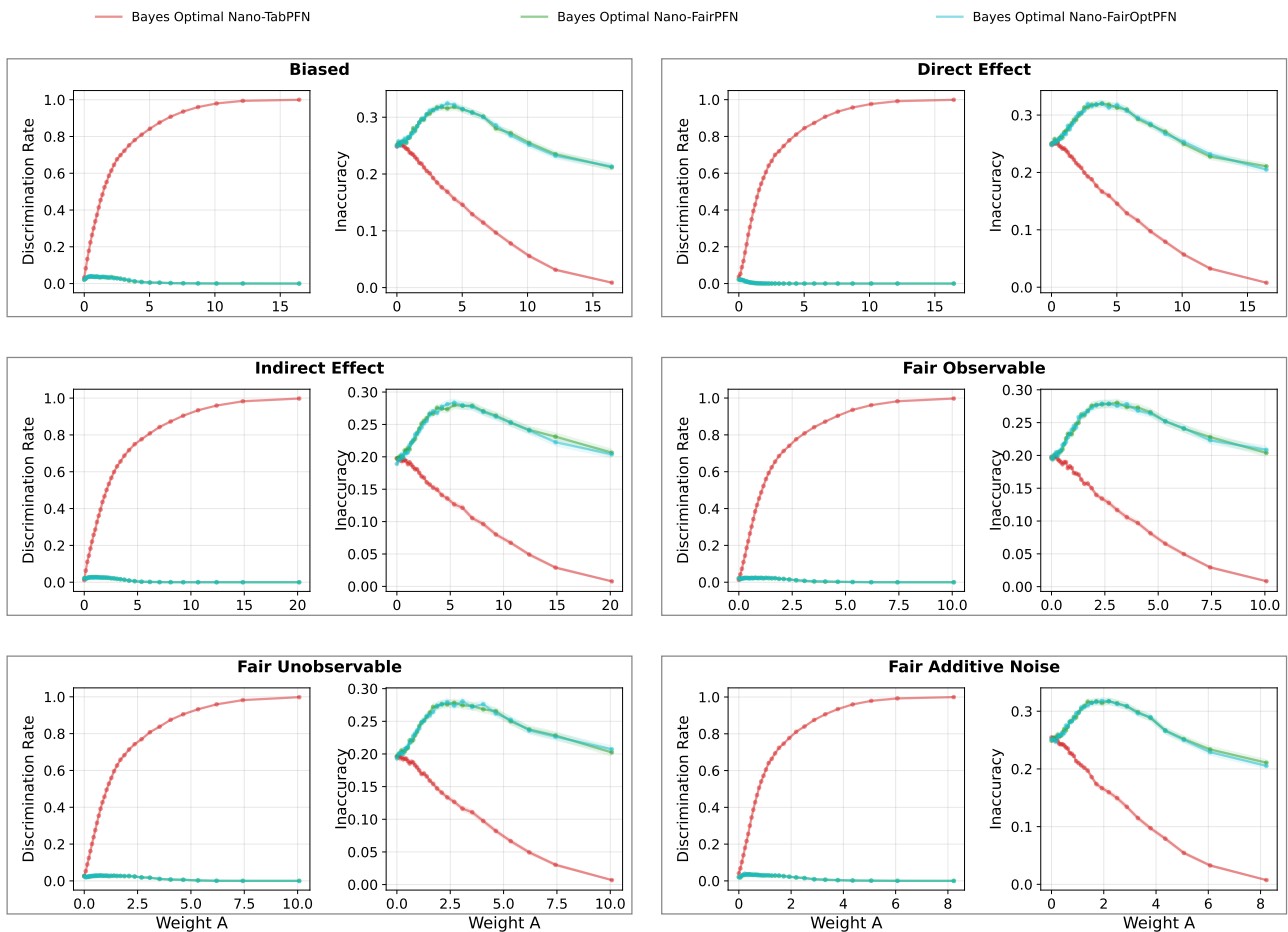

*Figure 9.* Additional results for $P(A = 1) = 0.2$. Because pre-training and evaluation of PFNs is expensive, we used these Bayes Optimal curves to show what the curves of learned PFNs would look like for this setting (had the learned PFNs approximated PPD with zero approximation error). When $\Theta$ is parametrized with a $P(A = 1) \leq 0.5$, then $y_f = y_f^\star$. In these hypothesis spaces FairPFN and FairOptPFN are identical.

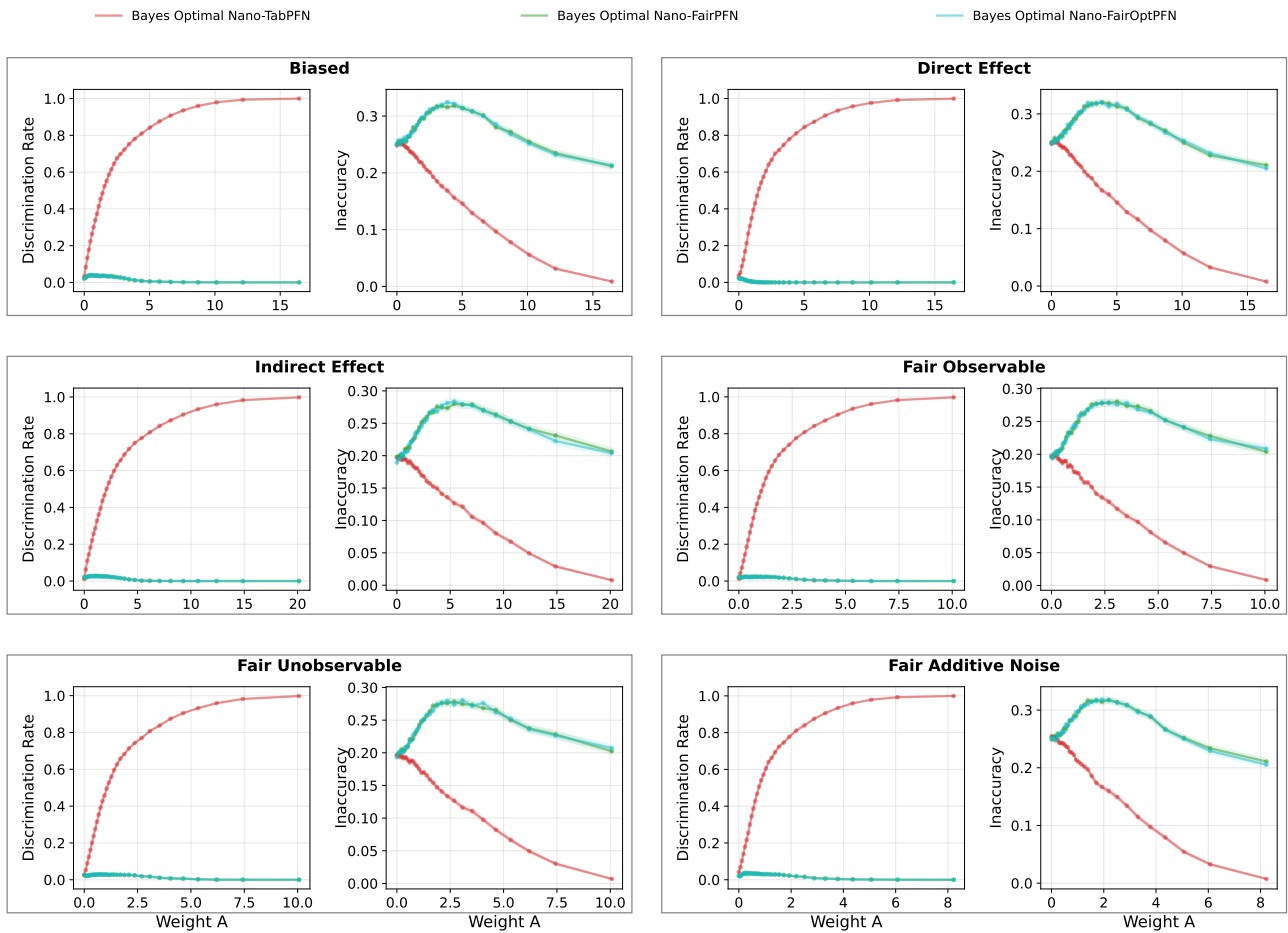

*Figure 10.* Specialized results for $P(A = 1) = 0.2$. Here the DAG is fixed, and only the weights exiting A are sampled randomly.

