# OpenReview forum: "FairOpt-PFN: Amortized Counterfactual Fairness with Optimal Fair Targets"
_ICML.cc/2026/Workshop/FMSD — FMSD @ ICML 2026 Poster_

### Official Review · Reviewer_7pGU · 2026-05-20
**A promising approach to enhancing fairness–utility trade-offs in pre-trained TFMs, but the experimental evaluation is somewhat limited.**

**Rating:** 6
**Confidence:** 4

**Review:**

### **Summary**

The paper introduces FairOpt-PFN, a tabular foundation model for causal fairness designed to improve the fairness–utility tradeoff relative to FairPFN. It argues that FairPFN’s training procedure, which defines fair target labels from a causal graph by removing paths from the protected attribute, can lead to suboptimal predictive accuracy. FairOpt-PFN instead pre-trains on a “fair” target defined as the Bayes-optimal decision rule given the conditional distribution over the non-descendants of the protected attribute. Experiments on several SCM-based benchmarks show that this choice yields a better fairness–utility tradeoff, and highlight that the pre-training target definition can strongly influence downstream utility and fairness outcomes.

### **Strengths**

S1- The paper tackles an important problem. The observation that the choice of the pre-training target label have important impact on the fairness–utility tradeoff is a novel and useful contribution.

S2- The Bayes-optimal fair target is clearly defined, well motivated, and technically sound.

S3- The empirical evaluation is thorough. It compares multiple target definitions (including the unfair ground truth and targets predicted by nanoPFN) and clearly illustrates their impact on the fairness–utility tradeoff.

### **Areas for Improvement**

1- The main weakness of the paper is that the evaluation is limited to synthetic SCM benchmarks. It is important to evaluate on real-world datasets to demonstrate that the approach generalizes beyond synthetic settings.

2- The evaluation of FairPFN with a lightweight backbone might also hide the impact of scaling. A larger model might achieve better utility, so it would be helpful to include experiments with a stronger backbone (or to justify why the chosen size is representative).

---

### Official Review · Reviewer_N2Ki · 2026-05-20
**Better conterfactual fairness for PFNs, synthetic evaluation only**

**Rating:** 6
**Confidence:** 2

**Review:**

# Summary
This paper improves the pre-training objective to boost counterfactual fairness of PFNs. The authors derive a Bayes-optimal fair target grounded in statistical decision theory, demonstrating improved fairness-utility trade-offs across controlled *synthetic* benchmarks. The evaluation is sound, but lacks any real world datasets.

# Strengths
- The proposed target is theoretically well-motivated. The fair pre-training framing as a constrained decision problem is coherent.
- The use of ATE-calibrated SCM benchmarks cleanly isolates approximation error from target choice.

# Weaknesses
- The optimal target is derived using the training-time marginal $p(A=1) = 0.8$, yet this quantity is treated as a fixed hyperparameter rather than something estimated from the deployment distribution (which makes sense from the pre-training perspective). As a result, the optimality guarantee does not transfer to test distributions where $p(A) \neq 0.8$, which undermines the theoretical motivation for the proposed target.
- Evaluation is limited to synthetic data and it is unclear whether the improvements would hold on real-world tabular fairness datasets (e.g. Adult, COMPAS).

# Suggestions
- Report results on a few real-world datasets to strengthen empirical claims
- Discuss or ablate sensitivity to the assumed marginal $p(A)$, e.g. by varying it across pre-training runs